# Newly Emerging Pest in China, *Rhynchaenus*
*maculosus* (Coleoptera: Curculionidae): Morphology and Molecular Identification with DNA Barcoding

**DOI:** 10.3390/insects12060568

**Published:** 2021-06-21

**Authors:** Rui-Sheng Yang, Ming-Yang Ni, Yu-Jian Gu, Jia-Sheng Xu, Ying Jin, Ji-Hui Zhang, Yong Wang, Li Qin

**Affiliations:** 1College of Bioscience and Biotechnology, Shenyang Agricultural University, Shenyang 110866, China; axiaozhuzai5@163.com (M.-Y.N.); g424335115@163.com (Y.-J.G.); yongwang216@163.com (Y.W.); 2School of Life and Environmental Sciences, Gannan Normal University, Ganzhou 341000, China; xjstis@163.com; 3Jilin Provincial Sericulture Institution, Agriculture Committee of Jilin Province, Jilin 132012, China; jin1975_ying@163.com (Y.J.); hui4238887@163.com (J.-H.Z.)

**Keywords:** leaf-mining pest, oak trees, forest pest, rapid identification, early monitoring

## Abstract

**Simple Summary:**

Accurate and rapid species identification is crucial in early monitoring and intervention of a pest, especially for a new and emerging pest. Here, the first detailed morphological features of *Rhynchaenus maculosus* at four life-cycle stages are reported for traditional species identification. Morphology-based identification is not only time-consuming but often inconclusive for closely related sibling species, juveniles, and even not feasible in the absence of key morphological characters. To overcome the limitations of traditional methods in species identification, DNA barcodes (mitochondrial cytochrome C oxidase subunit 1, CO1) were successfully used to identify the pupal and larval specimens as R. maculosus and allocate the undetermined morphospecies in GenBank into their possible taxa. The integration of molecular identification with the morphology-based identification proved to be effective in the accurate and rapid identification of R. maculosus.

**Abstract:**

The oak flea weevil, *Rhynchaenus*
*maculosus* Yang et Zhang 1991, is a newly emerging pest that severely damages oak (genus *Quercus*) in China. The first *R. maculosus* outbreak occurred in 2020 and caused spectacular damage to all oak forests in Jilin province, northeast China. The lack of key morphological characters complicates the identification of this native pest, especially in larva and pupa stages. This is problematic because quick and accurate species identification is crucial for early monitoring and intervention during outbreaks. Here, we provided the first detailed morphological description of *R. maculosus* at four life stages. Additionally, we used DNA barcodes from larva and pupa specimens collected from three remote locations for molecular identification. The average pairwise divergence of all sequences in this study was 0.51%, well below the 2% to 3% (K-2-parameter) threshold set for one species. All sample sequences matched the *R. maculosus* morphospecies (KX657706.1 and KX657707.1), with 99.23% to 100% (sequence identity, E value: 0.00) matching success. The tree based on barcodes placed the specimens into the *Rhynchaenus* group, and the phylogenetic relationship between 62 sequences (30 samples and 32 from GeneBank) had high congruence with the morphospecies taxa. The traditional DNA barcodes were successfully transformed into quick response codes with larger coding capacity for information storage. The results showed that DNA barcoding is reliable for *R. maculosus* identification. The integration of molecular and morphology-based methods contributes to accurate species identification of this newly emerging oak pest.

## 1. Introduction

The genus *Rhynchaenus* forms an economically significant pest group with a global distribution that includes the Northern Hemisphere, South Africa, Madagascar, and Australia [1]. It is the most speciose and structurally diverse genus of the Rhynchaeninae subfamily of Curculionidae. Members are commonly known as “flea weevils” and notable for their leaf-mining larval habits [1]. Over 70 and 40 *Rhynchaenus* species were reported in the former Soviet Union countries [2] and Japan [3], respectively. In contrast, only 11 species have been reported in China, including the oak flea weevil, *Rhynchaenus*
*maculosus* Yang et Zhang, *R. alni* L., *R*. sp. (Host plant: *Pterocarya*
*stenoptera*), *R*. sp. (Host plant: *Populus*
*euphratica, P. nigra*), *R. empopulifolis* Chen, *R. guliensis* Yang et Dai, *R*. *mutabilis* Boheman, *R. harunire* Morimoto, *R. stigma* Germar, *R. populi* Fabricius, and *R. mangiferae* Mashaee [2,4,5,6,7].

The oak flea weevil, *R. maculosus*, was first recorded as a new species in China and identified based on the general morphological features of adults collected from the Daxing’anling area, Heilongjiang province, in 1987 [4]. There were no additional reports of this species globally until late April to late May 2012, when damage symptoms, including mid-vein to margin leaf mines and blister-like blotches on leaf margins, were discovered at the Oak Germplasm Resource Bank (123°34′31′′E, 41°49′47′′N) and the oak garden for Chinese oak silkworm rearing (123°34′35′′E, 41°49′52′′N), Shenyang Agricultural University, eastern Shenyang, China. Because of its small body size and cryptic feeding behavior, the oak damage caused by *R. maculosus* did not immediately attract the researchers’ attention, which resulted in a gradual increase in pest population and a growth-depressing impact on trees. Despite the progressive aggravation of oak leaf mining, we did not collect adult weevils in the wild because of insufficient biological information until late May 2015 when we gradually understood the traits of pupal development and adult eclosion. Then, we collected twenty-one pupae in the blister-like chambers (pupa cell) on oak leaves sampled in the wild and took them to the laboratory for adult eclosion. These leaf miners were identified morphologically as *R. maculosus* based on the characters of the adult type specimens deposited in the Department of Forestry, Northeast Forestry University, Harbin [4]. The only available approach for *R. maculosus* species identification relies on the limited morphological features of adult weevils. Identifying this species in the larval and pupal stages is challenging because of the lack of reliable morphological characters, small size, and similarity with other leaf-mining species in appearance, which hinder further studies on this pest.

Molecular markers have greatly enhanced the efficiency and accuracy of insect species identification [8,9,10,11] and are widely used by taxonomists to overcome the limitations of traditional morphology-based methods [12]. A short standardized fragment (600–800 bp) of the mitochondrial cytochrome C oxidase subunit 1 (*CO1*) gene has become a key tool to discriminate insect species rapidly and precisely [13,14,15,16,17], especially when morphological identification is intricate and problematic [18]. The *CO1* gene fragment has been widely used as a DNA barcode and as a species tag for all animal taxa [19].

The objectives of this study were to provide a detailed morphological description of the four developmental stages for traditional species identification of *R. maculosus*, verify the effectiveness of DNA barcode in species identification, and eventually establish molecular alternatives for rapid identification of this newly emerging and native pest in China.

## 2. Materials and Methods

### 2.1. Insect Collection

*Rhynchaenus**maculosus* adult, egg, larva, and pupa individuals used for morphological analysis were collected from 23 April to 31 May 2019, from oak trees (*Quercus*
*wutaishansea* Mary) at the Oak Germplasm Resource Bank (123°34′31′′E, 41°49′47′′N), Shenyang Agricultural University, located in eastern Shenyang, China. Adults were collected using sticky traps around the trunk combined with hand-collection on the oak buds from 23–25 April 2019. Eggs were collected by hand in the wild from 23–30 April 2019 and brought to the laboratory for morphological analysis. Larvae were collected by hand in the blister-like chambers near the leaf margins from 3–18 May 2019. Pupae were collected in the blister-like chambers (pupa cell) on the leaf margins from 23–31 May 2019. The dimensions of 20 individuals at each stage were measured under a stereoscope microscope. For adults, the body shape, length, width, and snout length were measured and recorded. The color and structure of appendages were recorded. After measurements and records, weevil adults were dissected for sexual determination based on the genitalia. For both eggs and larvae, the shape, size, and color were observed and recorded. For pupae, the size, shape, color, and appendages were measured and recorded.

Five larval and pupal individuals were each collected by hand in the blister-like chambers on oak leaves from three remote locations in Northeast China from 14 May to 7 June 2019 (Table 1) to provide DNA barcodes for species identification. All 30 samples were stored at −20 °C in 95% ethanol before DNA extraction.

### 2.2. Genomic DNA Extraction and Amplification

Genomic DNA was extracted from single individuals using the Universal Genomic DNA Extraction Kit (TaKaRa Biotechnology (Dalian) Co., Ltd., Shiga, Japan). The DNA was stored at −20 °C.

The mitochondrial *CO1* barcodingregion was amplified in a T100™ gradient Thermal Cycler (Bio-Rad Inc., Hercules, CA, USA) using the *CO1* invertebrates universal primers: forward, 5’-GGTCAACAAATCATAAAGATATTGG-3’, and reverse, 5’-TAAACTTCAGGGTGACCAAAAAATCA-3’ [20]. PCR was performed in a 35 μL reaction system containing 2 μL of DNA template (11 μg/mL), 1μL of each of the forward and reverse primers (0.3 μmol/L), 0.7 μL of dNTPs, 0.8 μL of Mighty Amp DNA polymerase (0.11 U/μL), 3.5 μL of 2× MightyAmp Buffer, and 26 μL of dd H_2_O. The PCR protocol included a pre-denaturation step of 3 min at 94 °C, followed by 35 cycles of 30 s at 95 °C, 45 s at 50 °C, and 1 min at 72 °C, with a final extension of 7 min at 72 °C. Amplification was confirmed by electrophoresis on 0.8% agarose gels stained with ethidium bromide.

### 2.3. Sequencing, Identification, and Phylogenetic Analysis

PCR products were purified with a gel extraction kit (TaKaRaBiotechnology (Dalian)Co., LTD). Purified PCR products of each sample were sent to TaKaRa Biotechnology (Dalian)Co.,Ltd. and directly sequenced in both directions. The primers used for sequencing were the same as the primers used for PCR. Assembled sequences from forward and reverse reads were obtained using ChromasPro 2.1.8 (http://technelysium.com.au/wp/chromaspro/ accessed on 11 December 2020) and manually checked in the BioEdit Sequence Alignment Editor ver.7.2.5 [21]. Nucleotide compositional skew was used to describe the nucleotide composition of the sequences, with the relative value of A to T (AT-skew=(A–T)/(A+T)) [22]. BLAST searches were conducted in the NCBI GenBank database (https://blast.ncbi.nlm.nih.gov/Blast.cgi) to determine the identity between each sequence obtained here and those deposited in the database. The variable sites (parsim-informative sites and singletons) among the 30 specimen sequences in this study were determined using MEGA 6.0 [23]. The 62 sequences (32 from GenBankand 30 sequenced here) were further aligned using Clustal X [24]. The *CO1* sequences from our specimens were transformed into Quick Response (QR) codes (https://www.liantu.com/ accessed on 21 September 2020), and two-dimensional DNA barcode symbologies were used for information storage, recognition, and retrieval of DNA barcode sequences.

The final dataset of 62 sequences was trimmed to 649 bp. A Neighbor-Joining (NJ) tree was constructed based on the Kimura 2-parameter model [25] using MEGA 6.0 [23]. DNA barcode-based species identification relied on two aspects: (1) sample sequences with a mean divergence value ≤ 2–3% threshold [10] were considered to be the same species, and (2) sequence matches > 99% with the morphospecies in GenBank were used to diagnose the species identity of each specimen in the present study. The sequence divergence was calculated by MEGA 6.0 [23]. All sequences in this study were deposited in GenBank (Table 1). Sequence divergences were calculated using the maximum p-distances under the Kimura 2-parameter model [25]. 

## 3. Results

### 3.1. Morphology

The eggs were 0.55 ± 0.07 mm (mean ± SD) in length and 0.35 ± 0.05 mm (mean ± SD) in width at the middle, elliptically shaped with a smooth surface and narrowing ends abruptly rounded (Figure 1). The color gradually became darker, from creamy-white when newly oviposited to yellow before hatching.

Larvae, legless and creamy-white without setae, were dorso-ventrally flattened with sharply defined segments (Figure 2). The body consisted of a less sclerotized prognathous head with highly sclerotized mouthparts, three thoracic segments, and eight abdomen segments. From the head backward, the segments gradually increased in width until the mid-abdomen, where they narrowed gradually toward the anal tip. The mature larvae reached 5.78 ± 0.27 mm (mean ± SD) in length and 1.46 ± 0.21 mm (mean ± SD) in width at the 2nd–3rd abdomen segment, the widest part of the body.

Pupae, 4.59 ± 0.30 mm (mean ± SD) in length and 1.84 ± 0.20 mm (mean ± SD) in width at the broadest segment, were exarate in type and fusiform in shape, with two narrowing ends (Figure 3). The whole body was milky-white with two developed brown compound eyes at the beginning of pupation, gradually becoming dark brown when approaching the late pupa stage. At the mid pupa stage, the abdominal and metathoracic segments were yellowish-white in color, with one pair of short, fine, and light brown setae symmetrically on either side of the dorsal midline. The head, prothorax, and mesothorax had the same color as the abdomen, with one pair of long, thick, and brown setae symmetrically on either side of the dorsal central line. The forewings were yellowish-white, with shorter, sparse, and yellowish-white setae, and the hind wings were brown in color without setae at the mid stage. 

The adult body, oval in shape and light brown in color, was entirely covered with dense and yellow-gray hair (Figure 4a, b). There was high similarity in morphology between male and female adults. Females had an average length of 4.07 ± 0.22 mm (mean ± SD) and width of 1.94 ± 0.09 mm, while males measured 3.94 ± 0.21 mm (mean ± SD) and 1.81 ± 0.10 mm, respectively. The head was dorsally shaped like an isosceles triangle, with a significant sulcus along the midline on the bottom side, and two dark brown compound eyes conspicuously at either side of the top of the triangle (Figure 4b). The adult had a feebly curved snout between the two compound eyes, which was 1.01 ± 0.06 mm (mean ± SD), approximately as long as the total of head and prothorax (Figure 4a). The snout, dark brown in color and well-developed with sparse hair, was opisthognathous in type and invisible dorsally (Figure 4a, c). Antennae, covered with sensillum, light brown in color, were located on the snout and consisted of the scape, pedicel, and flagellum. The flagellum was nine-segmented, the first six segments of which had sparse sensilla and the top three segments were abruptly rounded with dense sensilla, definitely larger than the others, resulting in an oval shape (Figure 5). The pronotum, strongly developed, wastrapezium-like in shape with arched lateral edges and covered by dense and gray-brown hair. Along the longitudinal midline of the pronotum was a conspicuously concave sulcus wider than on head capsule (Figure 4b). Legs, with sparse gray-brown hair, were light brown. The femur of the fore- and mid-legs were armed with a single spine on the ventral side, and the hind-leg had a highly developed and triangle-like femur armed with a row of spines on the ventral side irregular in length (Figure 6). Elytra were slightly punctate, with longitudinal lines of yellow-gray hair densely covered on the back. The abdomen consisted of six highly sclerotized segments visible dorsally and was covered with short, sparse, and yellow-gray hair.

### 3.2. DNA Barcoding and Molecular Identification of R. maculosus

This study produced a final alignment of DNA barcoding sequences of 649 bp without gaps for all the 30 sequences. The average nucleotide composition was T 32.7%, C 19.4%, A 31.9%, and G 16.0%, heavily biased toward Ts and As (especially at the second codon site) (Table 2), as is expected in insect mitochondrial DNA [26]. The alignment of the 30 sequences revealed 13 variable sites, 10 of which were parsim-informative sites and three singletons (Figure 7). The average pairwise divergence among the 30 sequences was 0.51%, ranging from 0 to 0.72%, far less than the 2% to 3% threshold used to diagnose the individuals as the same species.

The original *CO1* gene DNA barcode fragment represents a single nucleotide sequence. The QR code proved to be superior in representing DNA barcode sequences efficiently [27]. In Figure 8, the different colors in the DNA barcode images on the left represent the different nucleotides and the numbers above the sequences represent their corresponding lengths, which can be used to provide clear sequence information. The two-dimensional codes on the right were successfully generated for the corresponding sequences on the left, which had a larger coding capacity for species information. By scanning the two-dimensional codes with the scanner, the original DNA barcode sequences along with the correlative information (e.g., Latin names, vouchers, taxon, and host plant) can be obtained quickly, making the species identification of this pest more convenient and quicker. 

The 32 homologous sequences obtained from the BLAST searches included 12 morphospecies: R. maculosus, Orchestes jota, O. avellanae, O. mixtus, Trichodes leucopsideus, Celetes decolor, Curculio venosus, C. sulcatulus, Acalyptus carpini, Curculionidae sp., Leiosoma deflexum, and Baris imberbis. The 30 sample sequences had the closest relationship with the morphospecies R. maculosus (KX657707.1 and KX657706.1), with a sequence match of 99.23–100% (sequence identity, E value: 0.00), which was supported by a minimum average divergence of 0.51%. Orchestes avellanae was the second closest morphospecies, as confirmed by an average divergence of 1.7%.

The clustering of the 62 *CO1* sequences in the NJ tree showed congruence with the morphospecies taxon. The analysis demonstrated that the 30 specimens clustered into one clade with the *R. maculosus* morphospecies in GenBank (Figure 9). This was strongly supported by an average distance of 0.5% (maximum distance: 0.9%) between samples and the *R. maculosus* sequence, the lowest value of all sequences obtained from GenBank. The remaining GenBank morphospecies sequences were separated into seven clades and clustered into their own clades according to the genera *Orchestes*, *Acalyptus*, *Leiosoma*, *Trichodes*, *Celetes*, *Curculio*, and *Baris* (Figure 9). This indicated that the *CO1* gene fragment could clearly distinguish between genera. Therefore, the 30 specimens were successfully diagnosed and identified as the same morphospecies, *R. maculosus*, indicating that *CO1* barcoding can be used effectively to identify individuals of this species.

## 4. Discussion

Four leaf-mining pest species that damage oak (genus *Quercus*) inhabit China, including *R. maculosus* (Coleoptera: Curculionidae), *R. guliensis* (Coleoptera: Curculionidae) [4,28], *Tischeria decidua* Wek (Lepidoptera: Lyonetiidae) [29,30], and *Acrocerops*
*brongniardella* Fabricius (Lepidoptera: Gracillariidae) [31]. Recently, *R. maculosus* has become an emergent oak pest, significantly increasing the rate of leaf damage in Liaoning province from 6.9% in 2016 to 15.4% in 2018 [32]. As predicted [32], the first outbreak of *R. maculosus* occurred in an oak plantation in Panshi, Jilin province, in 2020 (http://lytzt.jl.gov.cn/ztzl/jlsfjy/tpxw/202007/t20200727_7380510.html, accessed on 29 July 2020), increasing the urgency of further study of this pest. Accurate and rapid identification of *R. maculosus* at the species level is crucial for its further study. Aspects of species diversity, evolutionary relationship, and control strategy development of this pest should be taken into consideration in future studies.

Species identification based on morphology is not only time-consuming but often inconclusive for closely related species, and even not feasible in the absence of key morphological characters [33]. Conversely, DNA barcode is a quick and reliable molecular method for species identification and can successfully overcome those limitations [34,35,36,37]. In this study, barcodes based on the mitochondrial *CO1* gene successfully identified *R. maculosus* specimens in pupal and larval stages and assigned specimen sequences in GenBank to their appropriate taxon. Notably, four undetermined morphospecies deposited in GenBank (Curculionidae sp.: KR490172.1, Curculionidae sp.: JF889079.1, Curculionidae sp.: JF889075.1, Curculionidae sp.: JF889077.1) were allocated into the different clades (Figure 9), revealing their possible taxa. 

Although DNA barcoding is more accurate and quicker than morphology-based species identification, the large printout size and difficulty in data retrieval from the sequences limit its application [38]. The Quick Response (QR) code, a new format for information storage and retrieval, has the largest coding capacity and a relatively high compression ratio for information storage [27]. In our study, the pest DNA barcodes were transformed into QR codes and two-dimensional symbologies were obtained along with clear DNA barcode sequences. The two-dimensional DNA barcodes were more practical and encompassed more information than traditional formats, including the original DNA sequences, Latin names, vouchers, and the pest’s host species [27,38,39].

Several leaf-mining insects of the Curculionidae family (weevils) have become invasive pests with significant economic and ecological impacts [40]. Because *R. maculosus*, native in China, is a newly emerging pest in northeast China, early monitoring and intervention are crucial in preventing the potential establishment and spread to other areas of the country and even internationally. In general, the kind of monitoring for this pest relies on the development stages. For adults, morphology is enough for pest monitoring. For larvae and pupae, DNA barcoding is more effective for monitoring. In this study, the integration of DNA barcoding with morphology-based identification provided a powerful tool for the accurate and quick identification of *R. maculosus*, which is necessary for early pest forecasting.

## Figures and Tables

**Figure 1 insects-12-00568-f001:**
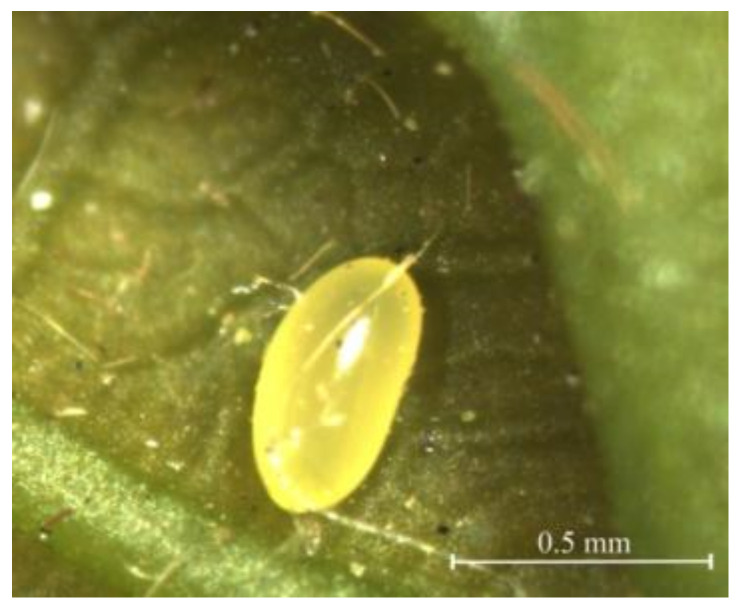
The egg of *Rhynchaenus*
*maculosus*. Shape: oval; coloration: smooth and bright yellow; development stage: late-stage.

**Figure 2 insects-12-00568-f002:**
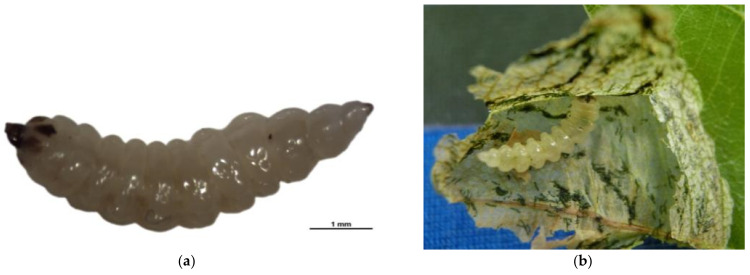
Larva of *Rhynchaenus*
*maculosus*. (**a**) Photograph taken in laboratory; (**b**) photograph taken in blister-like chamber on oak leaf. Both (**a**) and (**b**) show a creamy-white, legless body without setae, the sharply defined segments, highly sclerotized mouthparts, and less sclerotized prognathous head.

**Figure 3 insects-12-00568-f003:**
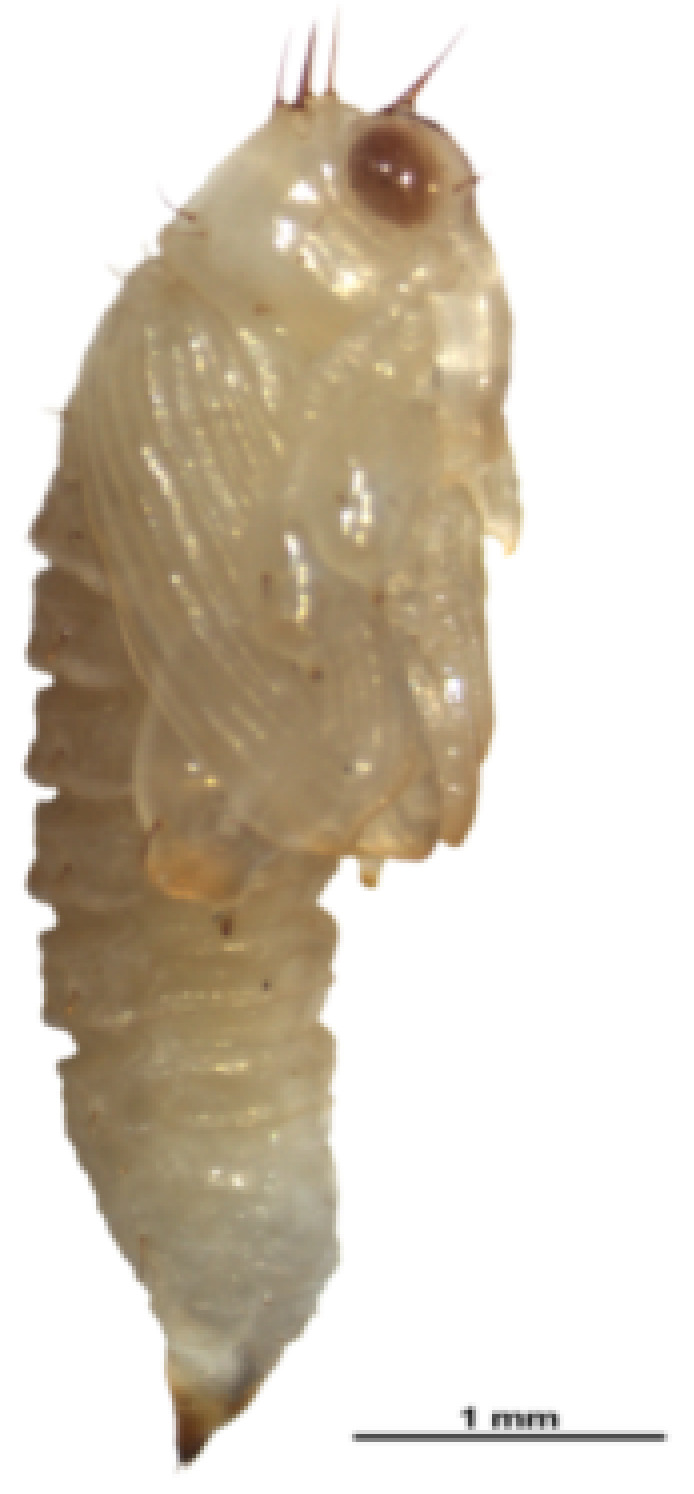
Early pupa of *Rhynchaenus*
*maculosus*. Exarate in type, fusiform in shape, and creamy-white body with sparse setae.

**Figure 4 insects-12-00568-f004:**
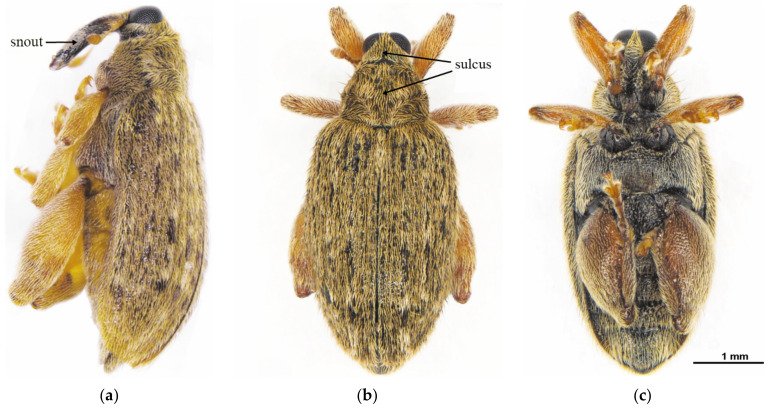
Adult of *Rhynchaenus maculosus*. (**a**) Lateral view; (**b**) dorsal view; (**c**) ventral view.

**Figure 5 insects-12-00568-f005:**
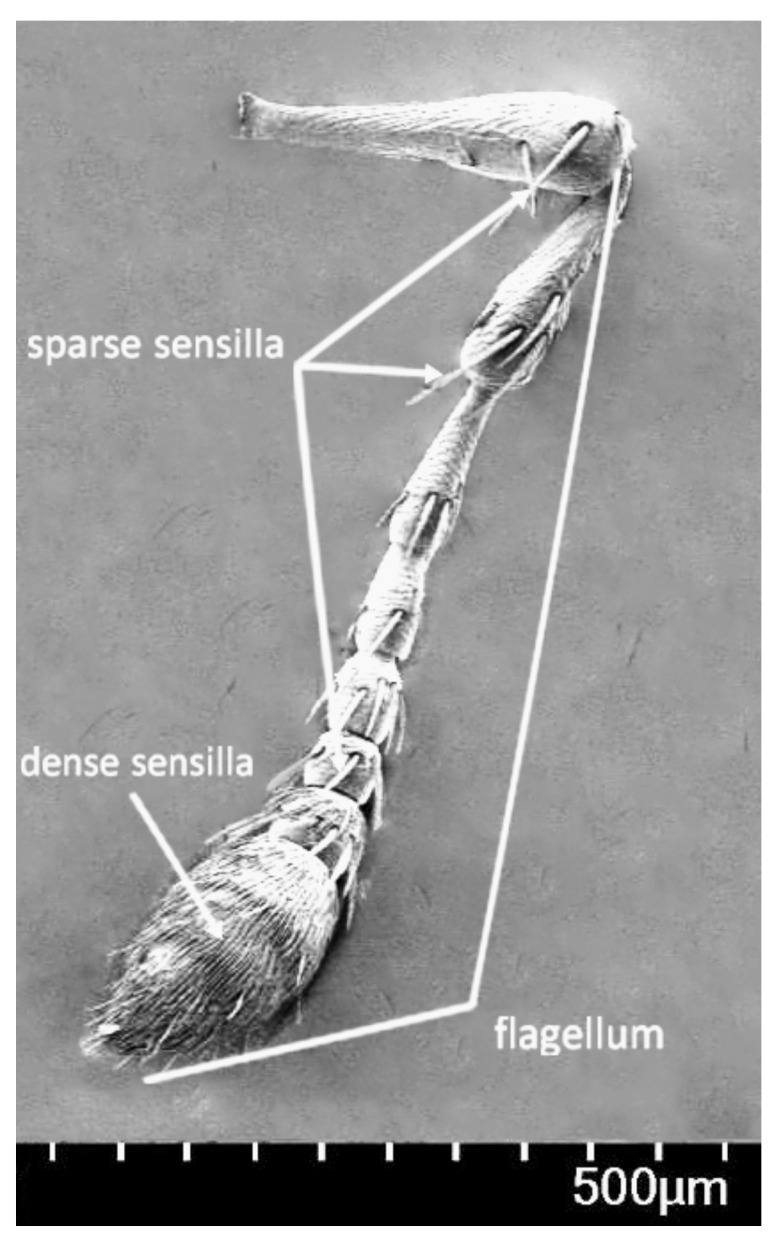
Antenna of *Rhynchaenus*
*maculosus* adult.

**Figure 6 insects-12-00568-f006:**
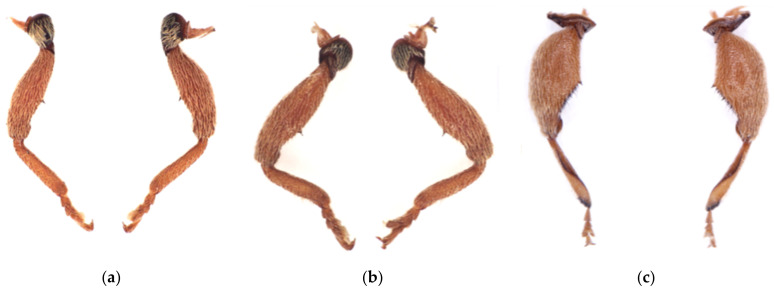
Legs of *Rhynchaenus*
*maculosus*. (**a**) Forelegs with a single spine on the ventral side of the femur; (**b**) mid-legs with a single spine on the ventral side of the femur; (**c**) hindlegs with a developed femur armed with a row of spines on the ventral side.

**Figure 7 insects-12-00568-f007:**
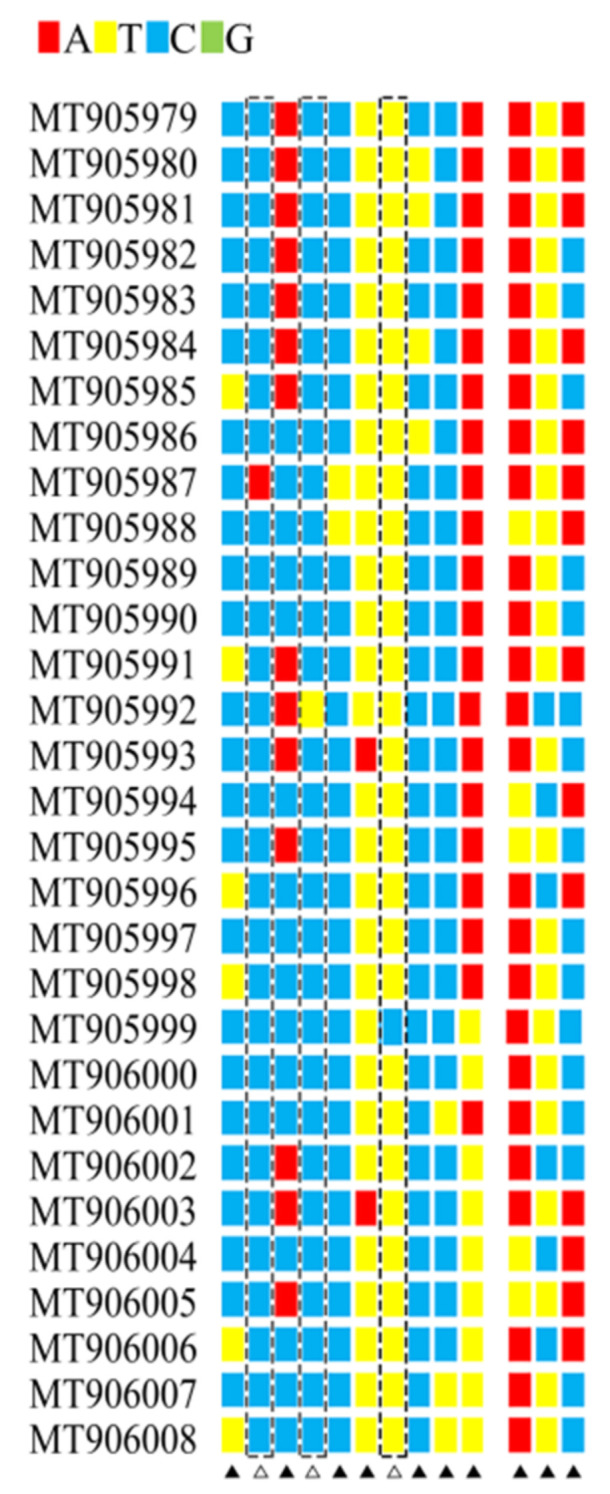
Variable sites of the 30 DNA barcode sequences. ∆ Singleton sites ▲ Parsim informative sites.

**Figure 8 insects-12-00568-f008:**
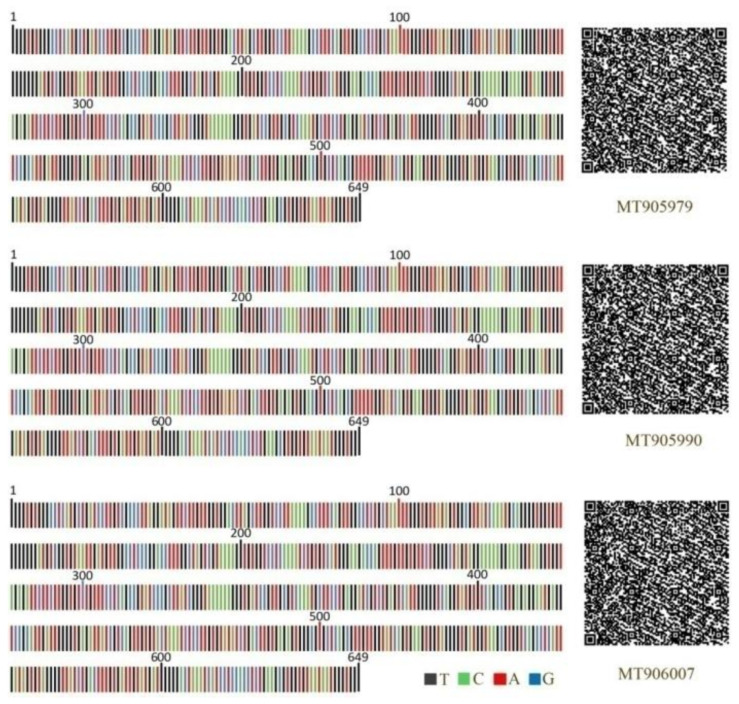
DNA barcodes and corresponding two-dimensional barcodes of partial specimens (MT905979, MT905990, MT90007) in this study.

**Figure 9 insects-12-00568-f009:**
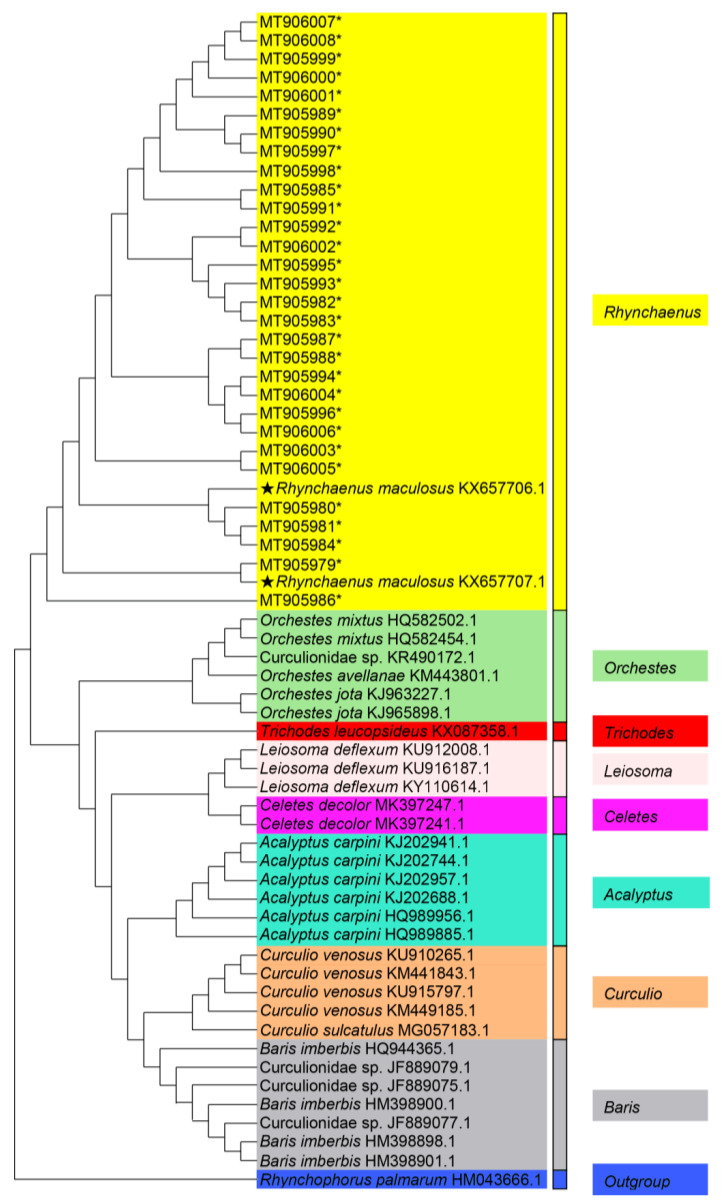
The neighbor-joining tree based on barcode sequences from specimens in the study and morphospecies in the GenBank database. * Specimens in the present study; ★ morphospecies of *Rhynchaenus*
*maculosus* already deposited in GenBank; color blocks indicate different clades.

**Table 1 insects-12-00568-t001:** Detailed information for the 30 specimens used for molecular identification.

Location/Province	Stage	Longitude/ Latitude *	Collection Date	Voucher	Host Plant	GenBank
Shenyang/Liaoning	Larva	123°34′ E/ 41°49′ N	14 May 2019	ZTX-L-SY01 ~ ZTX-L-SY05	* Querus wutaishansea *	MT905979 ~ MT905983
	Pupa	123°34′ E/ 41°49′ N	25 May 2019	ZTX-P-SY01 ~ ZTX-P-SY05		MT905984 ~ MT905988
Yongji/Jilin	Larva	126°30′E/ 43°39′N	28 May 2019	ZTX-L-YJ01 ~ ZTX-L-YJ05	* Querus mongolica *	MT905989~ MT905993
	pupa	126°30′E/ 43°39′N	4 June 2019	ZTX-P-YJ01 ~ ZTX-P-YJ05		MT905994 ~ MT905998
Jiamusi/Heilongjiang	Larva	130°22′ E/ 46°48′ N	31 May 2019	ZTX-L-JMS01 ~ ZTX-L-JMS05	* Querus mongolica *	MT905999~ MT906003
	pupa	130°22′ E/ 46°48′ N	7 June 2019	ZTX-P-JMS01 ~ ZTX-P-JMS05		MT906004 ~ MT906008

* Determined by Beidou Navigation Satellite System.

**Table 2 insects-12-00568-t002:** Nucleotide composition of DNA barcode sequences from the 30 specimens collected in this study.

Codon Site	T/%	C/%	A/%	G/%	A+T/%	G+C/%	AT Skew	GC Skew
1st site	41.4	25.9	15.1	17.6	56.5	43.5	−0.4655	−0.1908
2nd site	33.5	14.7	49.5	2.3	83.0	17.0	0.1928	−0.7294
3rd site	23.1	17.6	31.1	28.2	54.2	45.8	0.1476	0.2314
Average	32.7	19.4	31.9	16.0	64.6	35.4	−0.0124	−0.0960

## Data Availability

The insect specimens are deposited in Liaoning Research Centre for Insect Resource Engineering and Technology, Shenyang Agricultural University. All the raw barcode sequences in this study are available in the GenBank under accession number MT905979–MT906008.

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
