# Peer review of "Newly Emerging Pest in China, Rhynchaenusmaculosus (Coleoptera: Curculionidae): Morphology and Molecular Identification with DNA Barcoding"

_insects, 2021, doi:10.3390/insects12060568_

Round 1

Reviewer 1 Report

Thank you letting me read this interesting study!

I have only some minor comments.

Reviewer 2 Report

The Authors presented a diagnostic procedure based on DNA barcoding for an emerging pest of oak trees in China. Overall the topic is of interest, the manuscript is well written and the molecular procedure is almost fine to me, albeit more data must be analyzed. In fact, I think that the molecular analysis provided could be useful and worthy of publication in a full resarch article only if Authors include comparisons with closely-related species that are likely to occur in China. This would likely provide information of the taxonomic resolution of CO1 for this group of flea beetles and, indirectly, of the reliability of the analysis. I have the following comments:

-L46-47 and L48-54: I am not an expert of insect classification, but it seems that Rhynchaenus species are alternatively placed in Rhynchaeninae or Curculioninae subfamilies. Possibly some of them were renamed since the description reported in refs 2-7 cited . Please verify these aspects.

-L99-100: were insects preserved in ethanol at -20°C?

-L104: please, be more specific. Did you extracted from 1 or more legs/antennae/wings?

-L108: actually they are universal primers for invertebrates and not only insects

-L111-113: the volume is informative without details of the initial reagent concentrations. I suggest to only write the final concentration of each reagent in the 35ul reaction

-L125-126: why sequences were obtained from GenBank and not from BOLD database? BOLD provides COI sequences for taxonomically identified specimens, while GenBank not always? It would be great to verify whether deposited sequences can be retrived from BOLD and whether BLAST searches in BOLD provide similar phylogenetic results.

-It is not clear which taxonomic keys were used to identify Rhynchaenus maculosus. Additionally, because eleven species of Rhynchaenus occur in China, Authors must verify whether COI with Folmer primers offers adequate resolution for identification at the species level. Some sequences from closely related species (e.g. R. alni) are already available in BOLD, toghether with other European Rhynchaenus species (overall 179 records). If feasible, additional sequences for congeneric species can be possibly obtained from specimens deposited in museums or from specimens collected in the field.  Analyzing all toghether and reporting the results in this manuscript would likely make this ms a really interesting contribution on diagnostics of Rhynchaenus, not only for China but also for other Countries.      

Reviewer 3 Report

Please see my comments and suggestions in the annotated pdf.

Round 2

Reviewer 2 Report

My comments have been addressed. I have no further issues.

Author Response

Dear reviewer,

Thank you again for your constructive comments. We have revised the manuscript (Manuscript ID: insects-1243433) and sent the manuscript for your consideration as appendix to be published in Insects.

Reviewer 3 Report

The authors have improved on their previous version of this manuscript. Please see my comments and suggestions in the pdf file.
